# A Novel Class of Potent Anti-Tyrosinase Compounds with Antioxidant Activity, 2-(Substituted phenyl)-5-(trifluoromethyl)benzo[*d*]thiazoles: In Vitro and In Silico Insights

**DOI:** 10.3390/antiox11071375

**Published:** 2022-07-15

**Authors:** YeJi Hwang, Jieun Lee, Hee Jin Jung, Sultan Ullah, Jeongin Ko, Yeongmu Jeong, Yu Jung Park, Min Kyung Kang, Hwayoung Yun, Min-Soo Kim, Pusoon Chun, Hae Young Chung, Hyung Ryong Moon

**Affiliations:** 1Department of Manufacturing Pharmacy, College of Pharmacy, Pusan National University, Busan 46241, Korea; yjw4238@pusan.ac.kr (Y.H.); yijiun@pusan.ac.kr (J.L.); jungin8633@pusan.ac.kr (J.K.); dassabn@pusan.ac.kr (Y.J.); wjd9933@pusan.ac.kr (Y.J.P.); kmk87106@pusan.ac.kr (M.K.K.); hyun@pusan.ac.kr (H.Y.); minsookim@pusan.ac.kr (M.-S.K.); 2Department of Pharmacy, College of Pharmacy, Pusan National University, Busan 46241, Korea; hjjung2046@pusan.ac.kr (H.J.J.); hyjung@pusan.ac.kr (H.Y.C.); 3Department of Molecular Medicine, The Scripps Research Institute, Jupiter, FL 33458, USA; sullah@scripps.edu; 4College of Pharmacy and Inje Institute of Pharmaceutical Sciences and Research, Inje University, Gimhae 50834, Korea; pusoon@inje.ac.kr

**Keywords:** anti-tyrosinase, radical scavenging activity, anti-melanogenesis, tyrosinase glycosylation, 5-(trifluoromethyl)benzothiazole, B16F10 cells, kojic acid

## Abstract

Sixteen compounds bearing a benzothiazole moiety were synthesized as potential tyrosinase inhibitors and evaluated for mushroom tyrosinase inhibitory activity. The compound 4-(5-(trifluoromethyl)benzo[*d*]thiazol-2-yl)benzene-1,3-diol (compound **1b**) exhibited the highest tyrosinase activity inhibition, with an IC_50_ value of 0.2 ± 0.01 μM (a potency 55-fold greater than kojic acid). In silico results using mushroom tyrosinase and human tyrosinase showed that the 2,4-hydroxyl substituents on the phenyl ring of **1b** played an important role in the inhibition of both tyrosinases. Kinetic studies on mushroom tyrosinase indicated that **1b** is a competitive inhibitor of monophenolase and diphenolase, and this was supported by docking results. In B16F10 murine melanoma cells, **1a** and **1b** dose-dependently and significantly inhibited melanin production intracellularly, and melanin release into medium more strongly than kojic acid, and these effects were attributed to the inhibition of cellular tyrosinase. Furthermore, the inhibition of melanin production by **1b** was found to be partially due to the inhibition of tyrosinase glycosylation and the suppression of melanogenesis-associated genes. Compound **1c**, which has a catechol group, exhibited potent antioxidant activities against ROS, DPPH, and ABTS, and **1b** also had strong ROS and ABTS radical scavenging activities. These results suggest that 5-(trifluoromethyl)benzothiazole derivatives are promising anti-tyrosinase lead compounds with potent antioxidant effects.

## 1. Introduction

Melanin is a dark macromolecular pigment found in most organisms, including bacteria, fungi, insects, plants, invertebrates, and vertebrates. In mammals, there are two types of melanin, namely, eumelanin and pheomelanin, which are responsible for brown-to-black and yellow-to-red colorations, respectively. The primary function of melanin in animals is to protect the skin from ultraviolet rays, and to color skin, hair, feathers, and pupils [1]. In plants, melanin is closely related to the browning of fruits and vegetables. Melanin is secreted by melanocytes located in the basal epidermal layer. Melanosomes are organelles found in melanocytes, and biosynthesize melanin using a process referred to as melanogenesis, which involves a series of complex enzymatic and chemical processes. The functions of melanocytes are controlled by intrinsic factors, such as α-melanocyte-stimulating hormone (α-MSH), and extrinsic factors, such as chemicals and UV rays. Although melanin protects skin from UV radiation and by scavenging toxic chemicals and free radicals, its abnormal accumulation can cause hyperpigmentation-associated diseases, esthetic problems (e.g., freckles, melasma, and senile lentigines), and even skin cancer [2]. In addition, excessive melanin accumulation in crops makes maintaining the qualities of harvested fruits and vegetables difficult.

Tyrosinase (EC 1.14.18.1) is a copper-containing, multifunctional, glycosylated phenol oxidase (a polyphenolase), and has received considerable attention as an anti-melanogenic target because it is the enzyme responsible for the rate-determining step of melanogenesis [3], and is widely distributed, from bacteria to plants and mammals [4]. There are differences between mammalian tyrosinases and mushroom tyrosinase. Mushroom tyrosinase is a soluble tetrameric enzyme found in the cytoplasm, whereas mammalian tyrosinases, including human tyrosinase, are a glycosylated monomeric enzymes anchored to the melanosome membrane [5]. Although mushroom tyrosinase is only 22–24% identical to mammalian tyrosinases in the sequence range of 48–49%, most tyrosinase inhibitors have been identified based on their anti-tyrosinase activity against mushroom tyrosinase due to the low cost and commercial availability of mushroom tyrosinase [5]. Melanin is synthesized from l-tyrosine, which is oxidized to l-dopa by the monophenolase activity of tyrosinase. Subsequently, tyrosinase oxidizes l-dopa to l-*o*-dopaquinone, which, depending on the cellular environment, is converted to eumelanin or pheomelanin via different pathways, using enzymatic reactions involving tyrosinase-related protein-1 (TRP-1), tyrosinase-related protein-2 (TRP-2), and several chemical reactions [6].

Since tyrosinase is required for melanogenesis, it has become the most prominent and successful target of those seeking melanogenesis inhibitors that directly inhibit the catalytic activity of tyrosinase. Most commercially available skin-lightening agents are tyrosinase inhibitors, and numerous promising tyrosinase inhibitors have been identified for pharmaceutical, cosmeceutical, or agricultural purposes [7,8,9]. However, only a handful of compounds are used clinically due to insufficient efficacy or adverse effects, which include potential carcinogenicity. Arbutin, azelaic acid, hydroquinone, and kojic acid are used as skin-lightening agents for medical and cosmetic applications [10,11,12]. However, these agents have raised safety concerns, which include renal [13], immune system, bone marrow [14], and melanocyte [15] toxicities, and potential carcinogenicity [16]. Therefore, tyrosinase inhibitors are required that safely address unmet needs.

In an effort to discover novel tyrosinase inhibitors, we screened our in-house chemical library using mushroom tyrosinase [17], which has been widely used to identify potential fruit and vegetable anti-browning agents and skin-whitening agents, because it is commercially available, inexpensive, and provides reliable results. Preliminary screening using l-tyrosine as a substrate resulted in the identification of 2-(4-hydroxyphenyl)-5-(trifluoromethyl)benzo[*d*]thiazole (compound **1a**) as a potential tyrosinase inhibitor. Although the mushroom tyrosinase inhibitory efficacy of **1a** is lower than that of kojic acid, a representative tyrosinase inhibitor [18], **1a** at 10 and 50 µM effectively inhibited tyrosinase (Figure 1). According to our data on tyrosinase inhibitors [19,20,21,22], the numbers, types, and positions of substituents, especially hydroxyl groups, on the phenyl ring are strongly linked to tyrosinase inhibitory efficacy. Therefore, we altered the numbers, types, and positions of substituents on the phenyl ring of compound **1a** (Figure 1) and investigated their tyrosinase inhibitory activities.

Compound **1a** has a benzothiazole ring, which is analogous to other compounds with a 5-membered ring fused to benzene, such as indoles, benzimidazoles, benzofurans, and benzothiophenes, and these structures are widely used by medicinal chemists as scaffolds. Studies have shown that 2-arylbenzothiazole derivatives have diverse biological activities [23], such as antibacterial [24,25,26], anticancer [27,28,29,30], antioxidant [31], anti-tuberculosis [32], neuroprotective [33], hyaluronidase inhibitory [34], and antifungal properties [35]. In addition, the 2-arylbenzothiazole scaffold is found in various drugs, such as zopolrestat, pramipexole, ethoxazolamide, and entaluron.

We synthesized a series of 16 derivatives of compound **1a** and examined their tyrosinase inhibitory activities using mushroom and murine cellular tyrosinase, and their anti-melanogenic effects in murine B16F10 cells. In addition, we investigated the antioxidant effects of these derivatives using ROS (reactive oxygen species) and DPPH (2,2-diphenyl-1-picrylhydrazyl) and ABTS (2,2′-azino-bis(3-ethylbenzothiazoline-6-sulfonic acid)) radicals. The mode of mushroom tyrosinase inhibition by these 2-arylbenzothiazole derivatives was determined by kinetic studies, and the chemical structure primarily responsible for tyrosinase inhibition was investigated in silico.

## 2. Materials and Methods

### 2.1. Reagents

The kojic acid, trolox (6-hydroxy-2,5,7,8-tetramethylchroman-2-carboxylic acid), l-4-hydroxyphenylalanine (l-tyrosine), 2,2′-azino-bis(3-ethylbenzothiazoline-6-sulfonic acid) (ABTS), l-3,4-dihydroxyphenylalanine (l-dopa), phenylmethylsulfonyl fluoride (PMSF), α-melanocyte stimulating hormone (α-MSH), 3-isobutyl-1-methylxanthine (IBMX), 2,2-diphenyl-1-picrylhydrazyl (DPPH), 4-(1,1,3,3-tetramethylbutyl)phenylpolyethylene glycol (Triton^TM^ X-100), dimethyl sulfoxide (DMSO), 3-morpholinosydnonimine (SIN-1), potassium hydrogen phosphate, potassium dihydrogen phosphate, and mushroom tyrosinase were purchased from Sigma-Aldrich (St. Louis, MO, USA).

### 2.2. Chemistry

#### 2.2.1. General Methods

All chemicals were obtained from Thermo Fisher Scientific (Seoul, Korea), SEJIN CI Co. (Seoul, Korea), or Sigma-Aldrich (St. Louis, MO, USA). Reagents purchased were used without further purification. Solvents requiring anhydrous conditions were distilled over CaH_2_, or Na and benzophenone before use. Reactions were carried out in a nitrogen atmosphere and analyzed by thin layer chromatography (TLC) using pre-coated 60F_245_ plates purchased from Merck. Flash column chromatography using MP Silica 40–63, 60 Å was performed. Subsequently, ^1^H, ^13^C, and ^19^F NMR were measured at 500, 125, and 470 MHz, respectively. Mass data for low resolution were recorded in ESI modes (positive and negative) using an Expression CMS mass spectrometer (Advion Ithaca, NY, USA). CDCl_3_, CD_3_OD, and DMSO-*d*_6_ were used as solvents for NMR. All chemical shifts were measured in ppm (parts per million) versus residual solvent or deuterated peaks (*δ*_H_ 7.26, *δ*_H_ 3.31, and *δ*_H_ 2.48 for CDCl_3_, CD_3_OD, and DMSO-*d*_6_, respectively, and *δ*_C_ 77.0, *δ*_C_ 49.0, and *δ*_C_ 40.0 for CDCl_3_, CD_3_OD, and DMSO-*d*_6_, respectively). Coupling constants (*J*) are represented in hertz (Hz). The following abbreviations for ^1^H NMR were used: dd (doublet of doublets), brs (broad singlet), s (singlet), q (quartet), t (triplet), and d (doublet).

#### 2.2.2. General Synthetic Procedure for Compounds **1a**–**1l**

A solution of 2-amino-4-(trifluoromethyl)benzenethiol (100 mg, 0.44 mmol) and a substituted benzaldehyde (**a**–**l**, 1.0 equiv.) in methanol (5 mL) was stirred at room temperature for 4–15 h. After removing the solvent by evaporation, the resulting precipitate was filtered and washed with water, dichloromethane, and/or cold methanol to give compounds **1a** and **1c**–**1l** as solids in 28–68% yields. For **1b**, after solvent removal in vacuo, the resultant residue was purified by flash column chromatography, using hexane and ethyl acetate (4:1) as eluent, to give compound **1b** as a solid in 50% yield.

#### 2.2.3. Synthetic Procedure for Compound **1m**

A solution of 2-amino-4-(trifluoromethyl)benzenethiol (100 mg, 0.44 mmol) and 3,5-dimethyl-4-hydroxybenzaldehyde (66 mg, 0.44 mmol) in DMF (1 mL) containing Na_2_S_2_O_5_ (152 mg, 0.80 mmol) was heated at 80 °C for 14 h. After removing DMF in vacuo, methanol and water were added to the resulting liquid residue to precipitate the product. The precipitate generated was filtered and washed with water to give compound **1m** (72 mg, 51%) as a solid.

#### 2.2.4. Synthetic Procedure for Compound **1n**

A suspension of 2-amino-4-(trifluoromethyl)benzenethiol (100 mg, 0.44 mmol) and 3,5-dihydroxybenzaldehyde (61 mg, 0.44 mmol) in acetic acid (0.5 mL) containing sodium acetate (108 mg, 1.32 mmol) was refluxed for 7 h. Water was then added, and the resulting precipitate was filtered and washed with excess water. The filter cake was purified by flash column chromatography, using hexane and ethyl acetate (2:1) as eluent, to give compound **1n** (50 mg, 37%) as a solid.

#### 2.2.5. Synthetic Procedure for Compound **1o**

A suspension of 2-amino-4-(trifluoromethyl)benzenethiol (100 mg, 0.44 mmol) and 3,5-dibromo-4-hydroxybenzaldehyde (123 mg, 0.44 mmol) in DMF (3 mL) was heated for 13 h at 100 °C. After removing the DMF in vacuo, dichloromethane was added to the liquid residue to precipitate the product. The solid generated was filtered and washed with dichloromethane to give compound **1o** (54 mg, 28%).

#### 2.2.6. Synthetic Procedure for Compound **1p**

A suspension of 2-amino-4-(trifluoromethyl)benzenethiol (100 mg, 0.44 mmol) and 3-bromo-4-hydroxybenzaldehyde (87 mg, 0.43 mmol) in DMF (2 mL) and water (1 mL) containing Na_2_S_2_O_3_•5H_2_O (270 mg, 1.09 mmol) was heated for 16 h at 80 °C. After removing the solvent in vacuo, the resultant residue was partitioned between ethyl acetate and water, and the organic layer was dried over MgSO_4_, filtered, and concentrated under reduced pressure. After adding dichloromethane to precipitate the product, the resulting solid was filtered and washed with dichloromethane to afford compound **1p** (68 mg, 42%) as a solid.

#### 2.2.7. Characterization of compounds **1c**–**1p**

##### 4-(5-(Trifluoromethyl)benzo[d]thiazol-2-yl)phenol (**1a**)

Reaction time, 5.5 h; yield, 58%; melting point, 187.2–188.9 °C; ^1^H NMR (500 MHz, CD_3_OD) *δ* 8.18 (d, 1H, *J* = 1.0 Hz, 4′-H), 8.11 (d, 1H, *J* = 9.0 Hz, 7′-H), 7.95 (d, 2H, *J* = 8.5 Hz, 2-H, 6-H), 7.62 (dd, 1H, *J* = 1.0, 9.0 Hz, 6′-H), 6.92 (d, 2H, *J* = 8.5 Hz, 3-H, 5-H); ^13^C NMR (125 MHz, CD_3_OD) *δ* 171.1 (C2′), 161.2 (C4), 153.5 (C3a′), 138.2 (C7a′), 129.1 (C2, C6), 128.5 (q, *J* = 32.4 Hz, C5′), 124.4 (q, *J* = 269.6 Hz, *C*F_3_), 124.0 (C1), 122.4 (C7′), 120.7 (q, *J* = 3.5 Hz, C6′), 118.6 (q, *J* = 4.3 Hz, C4′), 115.7 (C3, C5); LRMS (ESI+) *m*/*z* 296 (M+H)^+^; (ESI−) *m*/*z* 294 (M−H)^−^.

##### 4-(5-(Trifluoromethyl)benzo[d]thiazol-2-yl)benzene-1,3-diol (**1b**)

Reaction time, 7 h; yield, 50.2%; melting point, 236.2–238.2 °C; ^1^H NMR (500 MHz, CD_3_OD) *δ* 8.10 (d, 1H, *J* = 1.0 Hz, 4′-H), 8.03 (d, 1H, *J* = 8.0 Hz, 7′-H), 7.58 (d, 1H, *J* = 8.5 Hz, 6-H), 7.56 (dd, 1H, *J* = 8.0, 1.0 Hz, 6′-H), 6.41 (dd, 1H, *J* = 8.5, 2.5 Hz, 5-H), 6.38 (d, 1H, *J* = 2.5 Hz, 3-H); ^13^C NMR (125 MHz, CD_3_OD) *δ* 171.1 (C2′), 162.5 (C4), 159.5 (C2), 151.6 (C3a′), 136.3 (C7a′), 129.9 (C6), 128.6 (q, *J* = 32.3 Hz, C5′), 124.3 (q, *J* = 269.4 Hz, *C*F_3_), 122.1 (C7′), 120.6 (q, *J* = 3.8 Hz, C6′), 117.7 (q, *J* = 3.8 Hz, C4′), 109.1, 108.3, 102.5; LRMS (ESI+) *m*/*z* 312 (M+H)^+^; LRMS (ESI−) *m*/*z* 310 (M−H)^−^.

##### 4-(5-(Trifluoromethyl)benzo[d]thiazol-2-yl)benzene-1,2-diol (**1c**)

Reaction time, 7.5 h; yield, 64.3%; melting point, 267.3–269.4 °C; ^1^H NMR (500 MHz, CD_3_OD) *δ* 8.14 (d, 1H, *J* = 1.0 Hz, 4′-H), 8.07 (d, 1H, *J* = 8.5 Hz, 7′-H), 7.59 (dd, 1H, *J* = 1.0, 8.5 Hz, 6′-H), 7.53 (d, 1H, *J* = 2.5 Hz, 2-H), 7.42 (dd, 1H, *J* = 2.5, 8.0 Hz, 6-H), 6.88 (d, 1H, *J* = 8.0 Hz, 5-H); ^13^C NMR (125 MHz, CD_3_OD) *δ* 171.4 (C2′), 153.3 (C3a′), 149.5 (C4), 145.7 (C3), 138.2 (C7a′), 128.5 (q, *J* = 32.5 Hz, C5′), 124.4 (q, *J* = 269.6 Hz, *C*F_3_), 124.4 (C1), 122.5 (C7′), 120.7 (q, *J* = 3.6 Hz, C6′), 120.1 (C6), 118.5 (q, *J* = 4.3 Hz, C4′), 115.4 (C5), 113.9 (C2); LRMS (ESI+) *m*/*z* 312 (M+H)^+^; LRMS (ESI−) *m*/*z* 310 (M−H)^−^.

##### 2-Methoxy-4-(5-(trifluoromethyl)benzo[d]thiazol-2-yl)phenol (**1d**)

Reaction time, 5 h; yield, 43.8%; melting point, 155.7–159.9 °C; ^1^H NMR (500 MHz, CD_3_OD) *δ* 8.18 (s, 1H, 4′-H), 8.14 (d, 1H, *J* = 8.5 Hz, 7′-H), 7.68 (d, 1H, *J* = 2.0 Hz, 2-H), 7.66 (d, 1H, *J* = 8.5 Hz, 6′-H), 7.55 (dd, 1H, *J* = 8.0, 2.0 Hz, 6-H), 6.93 (d, 1H, *J* = 8.0 Hz, 5-H), 3.97 (s, 3H, OCH_3_); ^13^C NMR (125 MHz, CD_3_OD) *δ* 171.9 (C2′), 151.5 (C4), 151.2 (C3a′), 148.2 (C3), 137.5 (C7a′), 128.9 (q, *J* = 32.3 Hz, C5′), 124.2 (q, *J* = 270.3 Hz, *C*F_3_), 123.1 (C1), 122.8 (C7′), 122.1 (C6), 121.1 (q, *J* = 2.9 Hz, C6′), 117.8 (q, *J* = 3.8 Hz, C4′), 115.5 (C5), 109.9 (C2), 55.2 (O*C*H_3_); ^19^F NMR (470 MHz, CD_3_OD) *δ* –55.81; LRMS (ESI−) *m*/*z* 324 (M−H)^−^.

##### 2-Ethoxy-4-(5-(trifluoromethyl)benzo[d]thiazol-2-yl)phenol (**1e**)

Reaction time, 6 h; yield, 28.8%; melting point, 117.0–117.8 °C; ^1^H NMR (500 MHz, CD_3_OD) *δ* 8.17 (d, 1H, *J* = 1.0 Hz, 4′-H), 8.10 (d, 1H, *J* = 8.5 Hz, 7′-H), 7.66 (d, 1H, *J* = 2.0 Hz, 2-H), 7.62 (dd, 1H, *J* = 8.5, 1.0 Hz, 6′-H), 7.52 (dd, 1H, *J* = 8.5, 2.0 Hz, 6-H), 6.92 (d, 1H, *J* = 8.5 Hz, 5-H), 4.19 (q, 2H, *J* = 7.0 Hz, OC*H_2_*CH_3_), 1.48 (t, 3H, *J* = 7.0 Hz, OCH_2_C*H_3_*); ^13^C NMR (125 MHz, CD_3_OD) *δ* 171.3 (C2′), 153.2 (C3a′), 150.7 (C4), 147.3 (C3), 138.2 (C7a′), 128.6 (q, *J* = 32.3 Hz, C5′), 124.3 (q, *J* = 270.3 Hz, *C*F_3_), 124.2 (C1), 122.5 (C7′), 121.6 (C6), 120.7 (q, *J* = 2.9 Hz, C6′), 118.4 (q, *J* = 3.8 Hz, C4′), 115.4 (C5), 111.0 (C2), 64.4 (O*C*H_2_), 13.6 (OCH_2_*C*H_3_); LRMS (ESI+) *m*/*z* 340 (M+H)^+^; LRMS (ESI−) *m*/*z* 338 (M−H)^−^.

##### 2-Methoxy-5-(5-(trifluoromethyl)benzo[d]thiazol-2-yl)phenol (**1f**)

Reaction time, 7 h; yield, 55.9%; melting point, 142.1–144.5 °C; ^1^H NMR (500 MHz, CD_3_OD) *δ* 8.18 (d, 1H, *J* = 1.0 Hz, 4′-H), 8.14 (d, 1H, *J* = 8.5 Hz, 7′-H), 7.66 (dd, 1H, *J* = 8.5, 1.0 Hz, 6′-H), 7.55 (dd, 1H, *J* = 8.5, 2.5 Hz, 6-H), 7.53 (d, 1H, *J* = 2.5 Hz, 2-H), 7.03 (d, 1H, *J* = 8.5 Hz, 5-H), 3.94 (s, 3H, OCH_3_); ^13^C NMR (125 MHz, CD_3_OD) *δ* 171.7 (C2′), 151.7 (C3a′), 151.5 (C4), 147.0 (C3), 137.5 (C7a′), 128.9 (q, *J* = 32.3 Hz, C5′), 124.4 (C1), 123.7 (q, *J* = 269.4 Hz, *C*F_3_), 122.8 (C7′), 121.2 (q, *J* = 3.8 Hz, C6′), 120.2 (C6), 118.0 (q, *J* = 3.8 Hz, C4′), 113.7 (C2), 111.4 (C5), 55.2 (O*C*H_3_); LRMS (ESI+) *m*/*z* 326 (M+H)^+^.

##### 2-(5-(Trifluoromethyl)benzo[d]thiazol-2-yl)phenol (**1g**)

Reaction time, 15 h; yield, 47.4%; melting point, 175.0–176.5 °C; ^1^H NMR (500 MHz, CDCl_3_) *δ* 12.11 (brs, 1H, OH), 8.22 (s, 1H, 4′-H), 7.97 (d, 1H, *J* = 8.5 Hz, 7′-H), 7.64 (d, 1H, *J* = 8.0 Hz, 6-H), 7.62 (d, 1H, *J* = 8.5 Hz, 6′-H), 7.40 (t, 1H, *J* = 8.0 Hz, 4-H), 7.10 (d, 1H, *J* = 8.0 Hz, 5-H), 6.95 (t, 1H, *J* = 8.0 Hz, 3-H); ^13^C NMR (125 MHz, CDCl_3_) *δ* 170.4 (C2′), 157.0 (C2), 150.5 (C3a′), 135.0 (C7a′), 132.5 (C4), 128.4 (q, *J* = 32.6 Hz, C5′), 127.5 (C6), 123.0 (q, *J* = 270.5 Hz, *C*F_3_), 121.1 (C7′), 120.9 (q, *J* = 3.0 Hz, C6′), 118.7 (C5), 118.3 (q, *J* = 3.8 Hz, C4′), 117.1 (C3), 115.2 (C1); LRMS (ESI+) *m*/*z* 326 (M+H)^+^; LRMS (ESI−) *m*/*z* 294 (M−H)^−^.

##### 2-(4-Methoxyphenyl)-5-(trifluoromethyl)benzo[d]thiazole (**1h**)

Reaction time, 10 h; yield, 67.7%; melting point, 128.8–130.1 °C; ^1^H NMR (500 MHz, CDCl_3_) *δ* 8.28 (d, 1H, *J* = 1.0 Hz, 4′-H), 8.03 (d, 2H, *J* = 8.5 Hz, 2-H, 6-H), 7.96 (d, 1H, *J* = 8.0 Hz, 7′-H), 7.58 (dd, 1H, *J* = 1.0, 8.0 Hz, 6′-H), 7.01 (d, 2H, *J* = 8.5 Hz, 3-H, 5-H), 3.89 (s, 3H, OCH_3_); ^13^C NMR (125 MHz, CDCl_3_) *δ* 168.9 (C2′), 161.4 (C4), 152.7 (C3a′), 137.2 (C7a′), 128.3 (C2, C6), 127.9 (q, *J* = 32.4 Hz, C5′), 124.7 (C1), 123.2 (q, *J* = 270.6 Hz, *C*F_3_), 121.0 (C7′), 120.1 (q, *J* = 3.1 Hz, C6′), 118.8 (q, *J* = 4.0 Hz, C4′), 113.5 (C3, C5), 54.5 (O*C*H_3_); LRMS (ESI+) *m*/*z* 310 (M+H)^+^.

##### 2-(2,4-Dimethoxyphenyl)-5-(trifluoromethyl)benzo[d]thiazole (**1i**)

Reaction time, 4 h; yield, 34.2%; melting point, 140.4–141.6 °C; ^1^H NMR (500 MHz, CDCl_3_) *δ* 8.53 (d, 1H, *J* = 9.0 Hz, 6-H), 8.33 (d, 1H, *J* = 1.0 Hz, 4′-H), 7.99 (d, 1H, *J* = 8.0 Hz, 7′-H), 7.57 (dd, 1H, *J* = 8.0, 1.0 Hz, 6′-H), 6.71 (dd, 1H, *J* = 9.0, 2.5 Hz, 5-H), 6.59 (d, 1H, *J* = 2.5 Hz, 3-H), 4.06 (s, 3H, OCH_3_), 3.91 (s, 3H, OCH_3_); ^13^C NMR (125 MHz, CDCl_3_) *δ* 165.4 (C2′), 163.6 (C4), 158.8 (C2), 151.3 (C3a′), 138.8 (C7a′), 131.0 (C6), 128.5 (q, *J* = 32.3 Hz, C5′), 124.4 (q, *J* = 271.3 Hz, *C*F_3_), 121.7 (C7′), 120.5 (q, *J* = 3.8 Hz, C6′), 119.2 (q, *J* = 4.8 Hz, C4′), 114.7 (C1), 106.2 (C5), 98.4 (C3), 55.7 (2-O*C*H_3_), 55.6 (4-O*C*H_3_); LRMS (ESI+) *m*/*z* 340 (M+H)^+^.

##### 2-(3,4-Dimethoxyphenyl)-5-(trifluoromethyl)benzo[d]thiazole (**1j**)

Reaction time, 7 h; yield, 37.6%; melting point, 120.4–121.6 °C; ^1^H NMR (500 MHz, CDCl_3_) *δ* 8.29 (s, 1H, 4′-H), 7.98 (d, 1H, *J* = 8.5 Hz, 7′-H), 7.72 (d, 1H, *J* = 2.0 Hz, 2-H), 7.61 (dd, 1H, *J* = 2.0, 8.5 Hz, 6-H), 7.59 (d, 1H, *J* = 8.5 Hz, 6′-H), 6.96 (d, 1H, *J* = 8.5 Hz, 5-H), 4.03 (s, 3H, OCH_3_), 3.97 (s, 3H, OCH_3_); ^13^C NMR (125 MHz, CDCl_3_) *δ* 170.0 (C2′), 153.4 (C3a′), 152.1 (C3), 149.4 (C4), 138.1 (C7a′), 128.9 (q, *J* = 33.3 Hz, C5′), 125.8 (C1), 124.2 (q, *J* = 270.3 Hz, *C*F_3_), 122.0 (C7′), 121.5 (C6), 121.2 (q, *J* = 3.9 Hz, C6′), 119.8 (q, *J* = 3.9 Hz, C4′), 111.0 (C2), 109.8 (C5), 56.1 (O*C*H_3_), 56.0 (O*C*H_3_); LRMS (ESI+) *m*/*z* 340 (M+H)^+^.

##### 5-(Trifluoromethyl)-2-(3,4,5-trimethoxyphenyl)benzo[d]thiazole (**1k**)

Reaction time, 7 h; yield, 65.4%; melting point, 132.5–134.0 °C; ^1^H NMR (500 MHz, CDCl_3_) *δ* 8.29 (s, 1H, 4′-H), 7.95 (d, 1H, *J* = 8.0 Hz, 7′-H), 7.58 (d, 1H, *J* = 8.0 Hz, 6′-H), 7.30 (s, 2H, 2-H, 6-H), 3.97 (s, 6H, 3-OCH_3_, 5-OCH_3_), 3.93 (s, 3H, 4-OCH_3_); ^13^C NMR (125 MHz, CDCl_3_) *δ* 169.8 (C2′), 153.6 (C3, C5), 153.5 (C3a′), 141.2 (C4), 138.3(C7a′), 129.0 (q, *J* = 33.3 Hz, C5′), 128.2 (C1), 124.2 (q, *J* = 270.4 Hz, *C*F_3_), 122.1 (C7′), 121.4 (q, *J* = 2.9 Hz, C6′), 120.1 (q, *J* = 3.9 Hz, C4′), 104.6 (C2, C6), 61.0 (4-O*C*H_3_), 56.3 (3-O*C*H_3_, 5-O*C*H_3_); LRMS (ESI+) *m*/*z* 370 (M+H)^+^, 425 (M+MeOH+Na)^+^.

##### 2,6-Dimethoxy-4-(5-(trifluoromethyl)benzo[d]thiazol-2-yl)phenol (**1l**)

Reaction time, 7 h; yield, 27.7%; melting point, 220.6–222.6 °C; ^1^H NMR (500 MHz, CD_3_OD) *δ* 8.15 (s, 1H, 4′-H), 8.10 (d, 1H, *J* = 8.5 Hz, 7′-H), 7.63 (d, 1H, *J* = 8.5 Hz, 6′-H), 7.30 (s, 2H, 2-H, 6-H), 3.93 (s, 6H, 3-OCH_3_, 5-OCH_3_); ^13^C NMR (125 MHz, CD_3_OD) *δ* 171.6 (C2′), 152.0 (C3a′), 148.2 (C3, C5), 140.0 (C4), 137.7 (C7a′), 128.8 (q, *J* = 32.3 Hz, C5′), 124.2 (q, *J* = 270.0 Hz, *C*F_3_), 122.7 (C1), 122.3 (C7′), 121.0 (q, *J* = 3.6 Hz, C6′), 118.0 (q, *J* = 4.0 Hz, C4′), 104.8 (C2, C6), 55.5 (2-O*C*H_3_, 6-O*C*H_3_); LRMS (ESI−) *m*/*z* 354 (M−H)^−^.

##### 2,6-Dimethyl-4-(5-(trifluoromethyl)benzo[d]thiazol-2-yl)phenol (**1m**)

Reaction time, 14 h; yield, 51.0%; ^1^H NMR (500 MHz, DMSO-*d*_6_) *δ* 9.11 (s, 1H, OH), 8.28 (d, 1H, *J* = 8.5 Hz, 7′-H), 8.24 (s, 1H, 4′-H), 7.67 (s, 2H, 2-H, 6-H), 7.66 (d, 1H, *J* = 8.5 Hz, 6′-H), 2.24 (s, 6H, 2×CH_3_); ^13^C NMR (125 MHz, DMSO-*d*_6_) *δ* 170.8 (C2′), 157.6 (C4), 153.8 (C3a′), 138.8 (C7a′), 128.2 (C2, C6), 127.8 (q, *J* = 31.9 Hz, C5′), 125.6 (C3, C5), 124.8 (q, *J* = 270.4 Hz, *C*F_3_), 123.9 (C1), 123.8 (C7′), 121.6 (q, *J* = 3.8 Hz, C6′), 119.2 (q, *J* = 4.1 Hz, C4′), 17.2 (2×*C*H_3_); LRMS (ESI+) *m*/*z* 324 (M+H)^+^; LRMS (ESI−) *m*/*z* 322 (M−H)^−^.

##### 5-(5-(Trifluoromethyl)benzo[d]thiazol-2-yl)benzene-1,3-diol (**1n**)

Reaction time, 6.5 h; yield, 37.3%; melting point, 256.7–260.6 °C; ^1^H NMR (500 MHz, CD_3_OD) *δ* 8.20 (d, 1H, *J* = 1.0 Hz, 4′-H), 8.10 (d, 1H, *J* = 8.5 Hz, 7′-H), 7.62 (dd, 1H, *J* = 8.5, 1.0 Hz, 6′-H), 7.01 (d, 2H, *J* = 2.5 Hz, 2-H, 6-H), 6.45 (t, 1H, *J* = 2.5 Hz, 4-H); ^13^C NMR (125 MHz, CD_3_OD) *δ* 171.2 (C2′), 159.1 (C3, C5), 153.2 (C3a′), 138.5 (C7a′), 134.2 (C1), 128.6 (q, *J* = 32.3 Hz, C5′), 124.3 (q, *J* = 269.9 Hz, *C*F_3_), 122.6 (C7′), 121.1 (q, *J* = 3.4 Hz, C6′), 119.2 (q, *J* = 4.1 Hz, C4′), 105.7 (C4), 105.7 (C2, C6); LRMS (ESI+) *m*/*z* 312 (M+H)^+^; LRMS (ESI−) *m*/*z* 310 (M−H)^−^.

##### 2,6-Dibromo-4-(5-(trifluoromethyl)benzo[d]thiazol-2-yl)phenol (**1o**)

Reaction time, 13 h; yield, 27.6%; ^1^H NMR (500 MHz, DMSO-*d*_6_) *δ* 10.65 (brs, 1H, OH), 8.37 (d, 1H, *J* = 8.5 Hz, 7′-H), 8.34 (s, 1H, 4′-H), 8.20 (s, 2H, 2-H, 6-H), 7.75 (d, 1H, *J* = 8.5 Hz, 6′-H); ^13^C NMR (125 MHz, DMSO-*d*_6_) *δ* 167.5 (C2′), 154.7 (C4), 153.4 (C3a′), 139.1 (C7a′), 131.5 (C2, C6), 128.1 (q, *J* = 31.3 Hz, C5′), 126.5 (C1), 124.7 (q, *J* = 270.4 Hz, *C*F_3_), 124.3 (C7′), 121.9 (q, *J* = 2.9 Hz, C6′), 119.9 (q, *J* = 3.8 Hz, C4′), 113.0 (C3, C5); LRMS (ESI−) *m*/*z* 450 (M−H)^−^, 452 (M−H+2)^−^, 454 (M−H+4)^−^.

##### 2-Bromo-4-(5-(trifluoromethyl)benzo[d]thiazol-2-yl)phenol (**1p**)

Reaction time, 16 h; yield, 41.6%; ^1^H NMR (500 MHz, DMSO-*d*_6_) *δ* 11.20 (brs, 1H, OH), 8.34 (d, 1H, *J* = 8.5 Hz, 7′-H), 8.31 (s, 1H, 4′-H), 8.20 (d, 1H, *J* = 2.0 Hz, 2-H), 7.92 (dd, 1H, *J* = 9.0, 2.0 Hz, 6-H), 7.72 (d, 1H, *J* = 8.5 Hz, 6′-H), 7.11 (d, 1H, *J* = 9.0 Hz, 5-H); ^13^C NMR (125 MHz, DMSO-*d*_6_) *δ* 169.0 (C2′), 158.0 (C4), 153.6 (C3a′), 138.9 (C7a′), 132.1 (C2), 129.0 (C6), 128.0 (q, *J* = 31.8 Hz, C5′), 125.3 (C1), 124.8 (q, *J* = 270.6 Hz, *C*F_3_), 124.2 (C7′), 121.7 (q, *J* = 3.4 Hz, C6′), 119.7 (*J* = 3.9 Hz, C4′), 117.4 (C5), 110.7 (C3); LRMS (ESI+) *m*/*z* 374 (M+H)^+^, 376 (M+H+2)^+^; LRMS (ESI−) *m*/*z* 372 (M−H)^−^, 374 (M−H+2)^−^.

#### 2.2.8. Synthetic Procedure for Compound **m**

A solution of 2,6-dimethylphenol (20.0 g, 163.72 mmol) and hexamethylenetetramine (7.8 g, 55.64 mmol) in acetic acid (40 mL) and water (20 mL) was refluxed for 14 h. After cooling to ambient temperature, volatiles were removed under reduced pressure. Ice water was added to the residue, and the precipitate generated was filtered and washed with water to obtain compound **m** as a solid. To extract compound **m** in filtrate, the filtrate was extracted using water and ethyl acetate, and the ethyl acetate layer was dried over anhydrous MgSO_4_, filtered, washed with water, and filtered to give compound **m** as a solid. The total yield of compound **m** obtained was 50.9% (12.522 g).

##### 4-Hydroxy-3,5-dimethylbenzaldehyde (**m**)

^1^H NMR (500 MHz, CDCl_3_) *δ* 9.78 (s, 1H, CHO), 7.53 (s, 2H, 2-H, 6-H), 5.76 (brs, 1H, OH), 2.31 (s, 6H, 2×CH_3_); ^13^C NMR (125 MHz, CDCl_3_) *δ* 191.7 (*C*HO), 158.5 (C4), 131.1 (C2, C6), 129.1 (C1), 124.0 (C3, C5), 15.9 (2×*C*H_3_); LRMS (ESI+) *m*/*z* 151 (M+H)^+^.

### 2.3. Tyrosinase Inhibition—Kinetic, In Silico, and In Vitro Studies

#### 2.3.1. Mushroom Tyrosinase Inhibition Assay

Mushroom tyrosinase inhibitory activity was determined as described previously [36], with minor modifications. l-Tyrosine was used as a substrate to determine enzyme activities. Briefly, an aqueous solution of mushroom tyrosinase (20 µL, 1000 units/mL) was added to each well of a 96-well microplate having a substrate mixture (170 µL) comprising l-tyrosine (345 μM) and sodium phosphate buffer (pH 6.5, 17.2 mM), and 10 µL of different concentrations of test compound in DMSO at concentrations determined by tyrosinase inhibitory activity or 10 µL kojic acid as a standard compound. Assay mixtures were incubated for 30 min at 37 °C, and amounts of dopachrome formed were determined by measuring absorbance at 475 nm using a microplate reader (VersaMax^TM^, Molecular Devices, Sunnyvale, CA, USA). These dose-dependent inhibition experiments were carried out three times using three to five different concentrations of test compounds to determine IC_50_ values. Log–linear curves and their equations were derived from the observed percentages of inhibition. IC_50_ values were defined as the concentrations that inhibited tyrosinase by 50%.

#### 2.3.2. Kinetic Studies on the Inhibition of Mushroom Tyrosinase by Compound **1b**

Lineweaver–Burk plots of compound **1b** were acquired in the presence of l-tyrosine or l-DOPA as substrates. In brief, 10 µL aliquots of DMSO containing compound **1b** (final concentrations: 0, 0.1, 0.2, or 0.4 µM in l-tyrosine, and 0, 1, 2, or 4 μM in l-DOPA) were added to the wells of a 96-well plate containing 170 µL of an aqueous solution (final concentrations: 1.0, 2.0, 4.0, 8.0, or 16.0 mM) of l-tyrosine or l-DOPA, sodium phosphate buffer (final concentration: 14.7 mM, pH 6.5), and 20 µL of mushroom tyrosinase (200 units/mL). Initial rates of dopachrome production were determined by measuring the increases in optical density at 475 nm (ΔOD_475_/min) using a microplate reader (VersaMax^TM^, Molecular Devices, Sunnyvale, CA, USA). Maximal velocity (V_max_) was obtained from Lineweaver–Burk plots obtained using five different l-tyrosine or l-DOPA concentrations. Modes of tyrosinase inhibition were determined using plot convergence points.

#### 2.3.3. In Silico Study of Interactions between Mushroom Tyrosinase and Compounds **1a** and **1b**, and Kojic Acid

The docking simulation study was conducted using the Schrödinger Suite (2021–2) with minor modifications, as previously described [20]. The *m*TYR (mushroom tyrosinase, PDB ID: 2Y9X, *Agaricus bisporus*) crystal structure was introduced from the Protein Data Bank (PDB) to Protein Preparation Wizard in Maestro 12.4. The *m*TYR crystal structure was processed, and unwanted protein chains were deleted. For the optimization of the structure, hydrogen atoms were added, water molecules more than 3 Å away from the enzyme were removed, and finally, the structure was minimized. The enzyme active site and the glide grid were assigned using the ligand (tropolone) binding site, determined using PDB and literature data [37,38,39]. The structures of compounds **1a** and **1b**, and kojic acid, were imported into the entry list of Maestro in CDXML format and prepared using LigPrep before ligand docking. The compounds were then docked to the glide grid of tyrosinase using the Glide task list **[40]**. Binding affinities and ligand–protein interactions were obtained using the extra precision (XP) glide method [41].

#### 2.3.4. In Silico Analysis of Molecular Interactions between Compounds **1a** and **1b**, and Kojic Acid, and a Human Tyrosinase Homology Model

The *h*TYR (human tyrosinase) homology model was generated using the SWISS-MODEL online server and the Schrödinger Suite (2020–2). The protein sequence of *h*TYR (P14679) was imported from the UniProt database, and the homology model was generated using the SWISS-MODEL online server based on the TRP-1 (PDB ID: 5M8Q) template. The homology model was further processed using the Schrödinger Suite and validated using Schrödinger prime (a homology modeling tool in the Schrödinger Suite). Compounds **1a** and **1b**, and kojic acid, were docked with the *h*TYR model using the protocols mentioned above for *m*TYR docking.

#### 2.3.5. Cell Culture 

B16F10 murine melanoma cells were obtained from the American Type Culture Collection (ATCC, Manassas, VA, USA). Phosphate buffer solution (PBS), Dulbecco’s modified Eagle’s medium (DMEM), streptomycin, penicillin, trypsin, and fetal bovine serum (FBS) were purchased from Gibco/Thermo Fisher Scientific (Carlsbad, CA, USA). B16F10 cells were cultured in a DMEM solution comprising 10% heat-inactivated FBS, penicillin (100 IU/mL), and streptomycin (100 µg/mL), at 37 °C in a humidified 5% CO_2_ atmosphere. Experiments for anti-melanogenesis activity, cell viability, and anti-tyrosinase activity were assayed on these cells in 96- or 6-well culture plates.

#### 2.3.6. B16F10 Cell Viability Assays

Cell viability assays for B16F10 were performed using the EZ-Cytox assay (EZ-3000, DoGenBio, Seoul, Korea) [42]. In brief, B16F10 cells were seeded in a 96-well plate at a density of 1 × 10^4^ cells/well, and cultured at 37 °C for 24 h in a humidified 5% CO_2_ atmosphere. The next day, B16F10 cells were exposed to six concentrations of **1a** or **1b** (0, 1, 2, 5, 10, or 20 μM), and incubated under the same conditions for 48 h. Then, the EZ-Cytox solution (10 μL) was added to each well, and the cells mixed with the EZ-Cytox solution were further incubated for 2 h. At 450 nm, optical densities were measured using a microplate reader (VersaMax^TM^, Molecular Devices, Sunnyvale, CA, USA). 

#### 2.3.7. Anti-Melanogenesis Activity Assay

The anti-melanogenic activities of **1a** and **1b** were determined using a standard melanin content assay [43], with minor changes. Briefly, B16F10 cells were seeded in the wells of a 6-well plate at a density of 1 × 10^5^ cells/well, and allowed to adhere to well bottoms over 24 h under the conditions used for cell culture (Section 2.3.5). Cells were exposed for 1 h to four different concentrations (0, 5, 10, or 20 µM) of **1a**, **1b**, or kojic acid (20 µM), stimulated with α-MSH (1 µM) plus IBMX (200 µM), and then incubated for 48 h in a humidified 5% CO_2_ atmosphere at 37 °C. Absorbances of culture media at 405 nm were used to determine extracellular melanin contents. 

Intracellular melanin contents were determined as follows: B16F10 cells were exposed to α-MSH plus IBMX treatment in the absence or presence of test compounds **1a**, **1b**, or kojic acid for 48 h. The cultured cells were rinsed with PBS twice and pellets were dissolved in 200 µL of 1N-NaOH solution containing 10% DMSO for 1 h at 60 °C. Cell lysates were then transferred to a 96-well plate, and the melanin absorbances were measured in aqueous DMSO at 405 nm using a microplate reader (VersaMax^TM^, Molecular Devices, Sunnyvale, CA, USA). 

#### 2.3.8. Evaluations of Anti-Tyrosinase Activities

Cellular tyrosinase activity was assessed by measuring the oxidation rate of l-DOPA as previously described [44], with minor modifications. In brief, B16F10 cells were seeded in a 6-well plate at 1 × 10^5^ cells/well, and allowed to adhere to well bottoms for 24 h, as described in Section 2.3.5. Cells were exposed for 1 h to **1a** or **1b** at 0, 5, 10, or 20 µM, or to kojic acid at 20 µM. Tyrosinase activity was induced by treatment of cells with IBMX (200 µM) and α-MSH (1 µM) together for 48 h (as described in Section 2.3.7). Cells were then rinsed twice with PBS, exposed to 100 µL of lysis buffer solution (90 µL of 50 mM phosphate buffer (pH 6.5), 5 µL of 2 mM PMSF, and 5 µL of 20% Triton X-100), and lysed at −80 °C for 30 min. After defrosting, cell lysates were transferred to microcentrifuge tubes and centrifuged at 12,000 rpm for 30 min at 4°C, and supernatants (80 µL) were mixed with 20 µL of l-DOPA (2 mg/mL) in a 96-well plate, and incubated for 10 min at 37 °C. Absorbances were measured at 475 nm using a microplate reader (VersaMax^TM^, Molecular Devices, Sunnyvale, CA, USA). 

#### 2.3.9. Protein Extraction and Western Blot Analysis

Five different concentrations of **1b** (0, 2, 5, 10, and 20 µM) were exposed to B16F10 cells for 48 h, and the cells were washed with cold PBS two times, harvested, lysed, and centrifuged at 4 °C (12,000 rpm for 10 min). Cell pellets were suspended in 10 mM Tris (pH 8.0) containing 1.5 mM MgCl_2_, 1 mM DTT, 0.1% NP-40, and protease inhibitors (GenDEPOT), incubated on ice for 15 min, and centrifuged at 12,000× *g* for 15 min at 4 °C. Supernatants (cytosolic fractions) and pellets were resuspended in 10 mM Tris (pH 8.0) containing 50 mM KCl, 100 mM NaCl, and protease inhibitor, incubated on ice for 30 min, and centrifuged at 12,000× *g* for 15 min at 4 °C. The resultant supernatants were nuclear fractions. The protein concentrations were determined by a Bicinchoninic Acid (BCA)™ Assay Kit (Pierce, Rockford, IL, USA). The lysed protein samples were mixed in loading buffer (0.2% bromophenol blue, 12.5 mM Tris-HCl, pH 6.8, 10% 2-mercaptoethanol, and 4% SDS) at a volume ratio of 1:4, and boiled for 10 min. Total protein of cell lysate was separated by SDS-PAGE using 9–10% acrylamide gels, and then transferred to polyvinylidene fluoride (PVDF) membranes (Millipore, Burlington, MA, USA) for 10 min at 25 V using a semi-dry transfer system (Bio-Rad Laboratories, Hercules, CA, USA). Following transfer, membranes were instantly placed in blocking buffer (5% non-fat dry milk in TBST (0.1% Tween-20, 100 mM NaCl, and 10 mM Tris (pH 7.5)) at room temperature for 1 h, followed by an overnight incubation with precise primary antibodies (1:500 to 1:2000 dilution) at 4 °C, washed for 10 min with TBST buffer (repeated three times), and subsequently incubated with appropriate horseradish peroxidase-conjugated secondary antibodies (anti-goat or anti-mouse antibodies (1:5000 dilution) for 1 h at room temperature. The specific antibodies for TFIIB (1:2000 dilution), β-actin (1:2000 dilution), MITF (1:500 dilution), and tyrosinase (1:1000 dilution) were purchased from Santa Cruz Biotechnology (Santa Cruz, CA, USA), and immunoreactive proteins were detected with the enhanced chemiluminescence (ECL) detection reagent (SuperSignal^®^ West Pico Chemiluminescent Substrate Kit, Advansta, San Jose, CA, USA). Western blot bands were visualized via a Davinch-Chemi™ imager (Davinch-K, Seoul, Korea). Quantification of Western blot data were analyzed by CS Analyzer 3.2 (Densitograph) image software (http://www.attokorea.co.kr, admitted on 21 May 2022).

#### 2.3.10. Total RNA Extraction and Quantitative Real-Time PCR (qRT-PCR)

Total RNAs from cells were extracted using the RiboEx Total RNA solution (GeneAll Biotechnology, Seoul, Korea). Cells in 6-well plates (0.5 mL, 1 × 10^5^ cells per well) were transferred to tubes, to which chloroform (0.1 mL) was added, and shaken vigorously for 30 s. Aqueous phases were transferred to fresh tubes, and an equal volume of isopropanol was added. Samples were then incubated for 15 min at 4 °C and centrifuged at 12,000× *g* for 15 min at 4 °C. After discarding the supernatants, RNA pellets were washed once with 0.5 mL of 75% ethanol, vortexed briefly, and centrifuged at 7500× *g* for 5 min at 4 °C. Pellets were dried for 10–15 min and dissolved in diethyl pyrocarbonate (DEPC)-treated water. Complementary DNA (cDNA) was synthesized from the total RNA (2 µg) using SuPrime Script RT Premix with random hexamer cDNA Synthesis Kit (GeNet Bio, Daejeon, Korea), in accordance with the manufacturers’ instructions. Quantitative qRT-PCR amplification was performed using SensiFAST^TM^ SYBR^®^ No-ROX dye (Bioline, London, UK) on the CFX Connect System (Bio-Rad Laboratories, Hercules, CA, USA). The primers specific to TRP-1, TRP-2, tyrosinase, or GAPDH were purchased from Bioneer Inc. (Daejeon, Korea). Relative gene expressions were assessed using GAPDH as the internal control. The primer sequences used in this study are shown in Table 1.

### 2.4. Antioxidant Activity 

#### 2.4.1. DPPH Radical Scavenging Activity Assay

The DPPH (2,2-diphenyl-1-picrylhydrazyl) scavenging activities of synthesized compounds **1a**–**1p** were investigated as previously described, with minor modifications [45]. A dimethyl sulfoxide solution (20 µL) containing 10 mM of each compound was mixed with DPPH methanol solution (0.2 mM, 180 µL) in the wells of a 96-well plate, and placed in the dark for 30 min at room temperature. At 517 nm, optical densities were measured using a microplate reader (VersaMax^TM^, Molecular Devices, Sunnyvale, CA, USA). The DPPH radical scavenging ability of the 2-arylbenzothiazole derivatives was compared with that of l-ascorbic acid, the positive control. All experiments were performed independently in triplicate. 

#### 2.4.2. ABTS Radical Scavenging Activity Assay

The ABTS (2,2-azino-bis(3-ethylbenzothiazoline-6-sulfonic acid) scavenging activities of compounds **1a**–**1p** were determined as previously described, with minor modifications [46]. In brief, ABTS radical solution was produced by adding 10 mL of 2.45 mM potassium persulfate to ABTS (7 mM in 10 mL distilled water), and storing the solution in the dark for 12–16 h at room temperature until the reaction was complete and absorbance remained constant. To study antioxidant activity, ABTS radical solution was diluted with water to an absorbance of 0.70 ± 0.02 at 734 nm; then, test compounds (10 µL, 100 µM) were added to 90 µL of ABTS free radical solution in the wells of a 96-well plate, and placed in the dark at room temperature for 2 min. At 734 nm, optical densities of the solutions were measured using a microplate reader (VersaMax^TM^, Molecular Devices, Sunnyvale, CA, USA). The ABTS radical scavenging capacities of the 16 2-arylbenzothiazole derivatives were compared with trolox. All experiments were performed independently in triplicate. 

#### 2.4.3. ROS Scavenging Evaluation

Intracellular ROS scavenging evaluation was performed as reported by Ali et al. [47] and Lebel and Bondy [48]. ROS generation was assessed using the ROS-sensitive fluorescence indicator DCFH-DA. Briefly, B16F10 cells were seeded in 96-well black plates (1 × 10^4^ cell/well), incubated for 24 h, treated with compounds **1a**–**1p** (20 µM in DMSO) for 2 h, stimulated with 10 µL of SIN-1 (100 µM in 50 mM sodium phosphate buffer, pH 7.4) for 1 h to induce ROS production, and then incubated with DCFH-DA (20 µM) for 30 min at 37 °C. Fluorescence was measured for 30 min at 5 min intervals at an excitation wavelength of 485 nm and emission wavelength of 535 nm, using a Berthold microplate reader (Berthold Technologies GmbH &Co., Wien, Austria).

### 2.5. Statistical Analysis 

One-way analysis of variance (ANOVA) followed by a Bonferroni post hoc test was used to determine the significances of intergroup differences. Analysis was achieved using GraphPad Prism 5 software (La Jolla, CA, USA), and results are shown as mean ± standard error of the mean (SEM). Two-sided *p*-values less than 0.05 were considered statistically significant.

## 3. Results and Discussion

### 3.1. Chemistry

Based on the structure of compound **1a**, which was found, via preliminary screening, to act as a tyrosinase inhibitor, and our accumulated SAR (structure–activity relationship) data on tyrosinase inhibitors, we decided to introduce various substituents, that is, a hydroxyl, bromo, alkoxyl, or methyl group to the *ortho*, *meta*, and/or *para* positions of the phenyl ring of **1a**. The syntheses of the compounds **1a**–**1p** were accomplished in one step, as illustrated in Figure 1. Commercially available 2-amino-4-(trifluoromethyl)benzenethiol was condensed with several benzaldehydes (**a**–**l**) in methanol in the absence of a catalyst to afford 2-(substituted phenyl)benzothiazole derivatives **1a**–**1l** in yields of 28–68%. However, compound **1p** could not be synthesized in the same manner as **1a**–**1l**. Synthesis was achieved by changing the reaction solvent to DMF and increasing the reaction temperature. The dibromophenyl-benzothiazole (compound **1o**) was produced by heating 2-amino-4-(trifluoromethyl)benzenethiol and 3,5-dibromo-4-hydroxybenzaldehyde (**o**) at 100 °C. An attempt was made to condense 2-amino-4-(trifluoromethyl)benzenethiol with 3-bromo-4-hydroxybenzaldehyde (**p**) in the presence of an oxidizing agent (Na_2_S_2_O_5_) to synthesize **1p** [49]; however, by mistake, the reaction was carried out in the presence of a reducing agent, Na_2_S_2_O_3_. Interestingly, the coupling of 2-amino-4-(trifluoromethyl)benzenethiol with 3-bromo-4-hydroxybenzaldehyde (**p**) in the presence of Na_2_S_2_O_3_ in DMF at 80 °C afforded the oxidized benzothiazole compound **1p**, instead of 2-bromo-4-(5-(trifluoromethyl)-2,3-dihydrobenzo[*d*]thiazol-2-yl)phenol (**1p′**). On the other hand, compound **1m** was synthesized by coupling 2-amino-4-(trifluoromethyl)benzenethiol with 4-hydroxy-3,5-dimethylbenzaldehyde (**m**), the latter of which was prepared from 2,6-dimethylphenol using Duff formylation conditions (hexamethylenetetramine and acetic acid) [50], in the presence of Na_2_S_2_O_5_ in DMF at 80 °C [49]. The corresponding 3,5-dihydroxyphenyl compound **1n** was obtained by refluxing 2-amino-4-(trifluoromethyl)benzenethiol and 3,5-dihydroxybenzaldehyde (**n**) in the presence of a weak acid and its sodium salt (e.g., CH_3_CO_2_H and NaOAc). The structures of compounds **1a**–**1p** were confirmed by ^1^H, ^19^F, and ^13^C NMR and mass spectroscopy. The assignments of ^1^H and ^13^C NMR peaks for all compounds were performed using the 2-(substituted phenyl)benzothiazole numbering system (see Figure 1). The C5′ carbon of the benzothiazole ring and the carbon of the trifluoromethyl group were observed as a quartet due to double-bond (^2^*J*_C,F_ ≈ 32 Hz) and single-bond (^1^*J*_C,F_ ≈ 270 Hz) couplings, respectively.

### 3.2. Mushroom Tyrosinase Inhibition and Log p Values

The inhibitory activities of the 16 derivatives were evaluated against mushroom tyrosinase using kojic acid and l-tyrosine as the positive control and substrate, respectively. All derivatives concentration-dependently inhibited mushroom tyrosinase. Table 2 summarizes their IC_50_ and log *p* values. 

Compound **1a** (IC_50_ = 54.2 μM) with a 4-hydroxyphenyl had fourfold less inhibitory activity than kojic acid (IC_50_ = 12.6 μM). Compounds **1h**, **1i**, and **1k** with no hydroxyl on the phenyl ring had IC_50_ values > 300 μM; in contrast, **1j** had an IC_50_ of 263.9 μM. Insertion of alkoxyl groups into the phenyl ring of **1a** greatly decreased tyrosinase inhibitory activity (IC_50_ = 291.1 μM for **1d**, and IC_50_ > 300 μM for **1e** and **1l**), whereas the insertion of two methyl groups decreased tyrosinase inhibition (IC_50_ = 128.9 μM for **1m**) by a factor of two. The introduction of one bromo group into the phenyl ring of **1a** halved the inhibitory efficacy (IC_50_ = 101.9 μM for **1p**), whereas the insertion of two bromo substituents (**1o**) reduced tyrosinase inhibitory to IC_50_ > 300 μM. Compound **1c**, with an additional hydroxyl substituent at position 3 of the phenyl ring of **1a**, also reduced tyrosinase inhibition (IC_50_ > 300 μM). However, **1b**, which had an additional hydroxyl group at position 2 of the phenyl ring of **1a**, increased anti-tyrosinase activity 240-fold (IC_50_ = 0.2 ± 0.01 μM), and was 55-fold more potent than kojic acid. The presence of a 2-hydroxyl group in the absence of the 4-hydroxyl group of **1a** resulted in an IC_50_ of > 300 μM (**1g**). Compound **1n**, with a 3,5-dihydroxyphenyl ring, exhibited weak inhibitory activity against mushroom tyrosinase (IC_50_ = 216.5 μM). As summarized in Figure 2, these results suggest that the 4-hydroxyl group on the phenyl ring contributes much to mushroom tyrosinase inhibition, and that tyrosinase inhibitory activity is further greatly increased when a 2-hydroxyl group is additionally introduced to the phenyl ring. However, the presence of the 2-hydroxyl group in isolation did not show tyrosinase inhibitory activity.

Compound **1b** exhibited dose-dependent tyrosinase inhibitory activities when l-tyrosine or l-DOPA were used as substrates (Figure 3), and showed stronger monophenolase and diphenolase inhibitory activities than kojic acid (Figure 3 and Table 3). The IC_50_ value of **1b** for l-DOPA was 1.7 ± 0.11 µM, which was lower than that of kojic acid (12.3 ± 2.74 µM). The mushroom tyrosinase inhibitory plot of **1b** demonstrates initial dose-dependent tyrosinase inhibitory activity, followed by a decrease in inhibitory activity, for l-tyrosine and l-DOPA at 2 and 10 µM, respectively. At concentrations less than 2 µM, **1b** inhibited monophenolase activity more than diphenolase activity, whereas at ≥10 µM, **1b** inhibited both to similar extents. These results suggest that, at low **1b** concentrations, monophenolase inhibition contributes more to inhibiting melanin production than diphenolase inhibition.

Melanocytes are located in the basal epidermal layer, and thus, potential tyrosinase inhibitors must be absorbed by the epidermis to exert their anti-melanogenic effects. Log *p* (partition coefficient in 1-octanol versus water) is an important metric because it reflects molecular lipophilicity in the neutral state, and is closely related to drug transport properties. ChemDraw Ultra Ver. 12.0 (Cambridge, MA 02140, USA) was utilized to obtain the log *p* values of the 16 derivatives. The log *p* values of all derivatives fell in the range 4.44–6.49 and were larger than that of kojic acid (−2.45), indicating all derivatives were more lipophilic than kojic acid. 

### 3.3. Enzyme Kinetics Mechanism Study

Mode of tyrosinase inhibition by **1b** was determined using Lineweaver–Burk plots in the presence of l-tyrosine or l-DOPA (Figure 4A,B). Lineweaver–Burk double reciprocal plots at four **1b** concentrations (0, 0.1, 0.2, and 0.4 µM for l-tyrosine, and 0, 1, 2, and 4 µM for l-DOPA) were all linear and intersected at one point on the y-axis, indicating **1b** acts as a competitive inhibitor of mushroom tyrosinase. K_m_ (Michaelis–Menten constant) values, calculated from Lineweaver–Burk double reciprocal plots, gradually increased with **1b** concentration from 1.77 mM at 0 µM to 18.79 mM at 0.4 µM in the presence of l-tyrosine, and from 1.78 mM at 0 µM to 5.20 mM at 4 µM in the presence of l-DOPA (Table 3). V_max_ (maximum velocity) values in the presence of l-tyrosine or l-DOPA were 1.01 × 10^−2^ and 4.97 × 10^−3^ mM/min, respectively, regardless of **1b** concentration. *K*_i_ (inhibition constant) values were obtained from Dixon plots in the presence of l-tyrosine or l-DOPA (Figure 4C,D). *K*_i_ values in the presence of l-tyrosine or l-DOPA were 3.76 × 10^−2^ and 2.45 µM, respectively (Table 3), indicating compound **1b** complexes better with mushroom tyrosinase acting as a monophenolase than as a diphenolase.

### 3.4. Docking Studies on Compounds ***1a*** and ***1b***

Since mushroom tyrosinase assays showed compounds **1a** and **1b** potently inhibited tyrosinase, we investigated their behaviors in the active site of tyrosinase. To determine their binding modes, we conducted docking studies using the Schrodinger Suite, release 2021–2. Results are shown in Figure 5 and Figure 6. 

#### 3.4.1. Binding Behaviors of Compounds **1a** and **1b** at the Active Site of Mushroom Tyrosinase 

As depicted in Figure 5, compounds **1a** and **1b** interacted with mushroom tyrosinase in the same way as kojic acid. In particular, the hydroxy-substituted phenyl rings of **1a** and **1b** interacted with the active site in a manner similar to kojic acid. Furthermore, the benzothiazole moieties of **1a** and **1b** also seemed to influence interactions with tyrosinase. 

Regarding the binding behavior of kojic acid, the hydroxyl group of its hydroxymethyl substituent coordinated with Cu401 ion at a distance of 2.36 Å, while the hydroxyl group of the 4-pyranone ring hydrogen-bonded with Met280 at a distance of 2.20 Å. In addition, the 4-pyranone ring of kojic acid formed a π–π stack with His263. The resulting docking score for kojic acid as determined by the Schrodinger Suite was −4.18 kcal/mol. As was observed for kojic acid, the phenolic ring of **1a** formed a π–π stacking interaction, but with His259 rather than His263. The docking score (−4.21 kcal/mol) of compound **1a** was similar to that of kojic acid. Interestingly, the 4-hydroxyl group of the phenyl ring of compound **1b** coordinated with Cu400 (distance 2.53 Å) and formed a salt bridge with Cu401 (distance of 2.31 Å). In addition, the 2-hydroxyl group of the phenyl ring in compound **1b** formed a 1.96 Å hydrogen bond with Glu256, and its phenyl ring formed a π–π stack with His259, as was observed for compound **1a**. The docking score of compound **1b** was −4.78 kcal/mol, which was higher than those of compound **1a** or kojic acid. Notably, the 5-(trifluoromethyl)benzo[*d*]thiazole moiety of compound **1b** was located in a hydrophobic environment, which we believe increased access to the active site of tyrosinase compared with kojic acid. 

#### 3.4.2. Binding Behaviors of Compounds **1a** and **1b** at the Active Site in the Human Tyrosinase Homology Model

Since the crystal structure of human tyrosinase has not been determined, we built a homology model based on TRP-1 to confirm the binding patterns of compounds **1a** and **1b** with human tyrosinase. As shown in Figure 6, both zinc ions coordinated with the hydroxymethyl group of kojic acid at distances of 2.29 Å and 2.10 Å, respectively. The hydroxyl group of the pyranone ring of kojic acid created a hydrogen bond at a distance of 2.04 Å with Ser375, while the 4-pyranone ring of kojic acid π–π stacked with His367. The recorded docking score for kojic acid was −4.45 kcal/mol. Thus, the binding pattern of kojic acid in the human tyrosinase homology model was similar to that observed for mushroom tyrosinase. The 4-hydroxyl group of the phenyl ring in compound **1a** formed one hydrogen bond at a distance of 2.13Å with Ser380, and coordinated with a zinc ion (Zn7) at an oxygen-to-Zn7 distance of 2.36 Å. In addition, the phenolic ring of **1a** π–π stacked with His367. The docking score of **1a** was −4.92 kcal/mol for **1a**. Compound **1b**, with an additional 2-hydroxyl group in the phenol ring of **1a**, generated two salt bridges at 2.12 Å and 2.33 Å distances with Zn6 and Zn7, respectively. In addition, the 2-hydroxyl of **1b** hydrogen bonded with Asn364 at a distance of 2.37 Å, and its phenyl ring π–π stacked with His367. These interactions resulted in the highest docking score for **1b** at −5.18 kcal/mol. These results indicate that **1a** and **1b** have the potential to inhibit human tyrosinase by binding to the active site of human tyrosinase.

The positive docking simulation results obtained for mushroom and human tyrosinase encouraged us to investigate the effects of **1a** and **1b** on cellular tyrosinase activities and melanogenesis in B16F10 murine melanoma cells.

### 3.5. Cytotoxic Effects of Compounds ***1a*** and ***1b***

As compounds **1a** and **1b** inhibited mushroom tyrosinase most, we investigated their cytotoxicity in B16F10 melanoma cells, and then their inhibitory effects on tyrosinase activity and melanogenesis in the same cells. An EZ-Cytox assay was used to evaluate cell viabilities. B16F10 cells were treated with different concentrations (0, 1, 2, 5, 10, and 20 μM) of **1a** and **1b** for 48 h in a humidified CO_2_ atmosphere. Optical densities of wells were measured using a microplate reader.

The effects of compounds **1a** and **1b** on cell viabilities are presented in Figure 7. Neither compound exhibited a significant cytotoxic effect at concentrations ≤20 μM. Therefore, B16F10 cell-based assays for tyrosinase activity and melanin production were conducted at ≤20 μM.

### 3.6. Inhibitory Effects of Compounds ***1a*** and ***1b*** on Extracellular Melanin Secretion by B16F10 Cells 

To investigate the inhibitory effects of **1a** and **1b** on melanin secretion into a medium, B16F10 cells were seeded in a 6-well culture plate and treated with 0, 5, 10, or 20 µM of the test compounds. One hour after plating, 1 µM of α-melanocyte-stimulating hormone (α-MSH) and 200 µM of 3-isobutyl-1-methylxanthine (IBMX) were added to increase tyrosinase activity, and after incubation for 48 h, melanin contents in the media were calculated by measuring the optical density of each well at 405 nm using a microplate reader. As the positive control, 20 µM of kojic acid was used.

Figure 8 shows the inhibitory effects of **1a** and **1b** on extracellular melanin release. Melanin contents in media increased by 106% when cells were co-treated with α-MSH and IBMX. Treatment with **1a** or **1b** concentration-dependently and significantly reduced melanin content increases in media induced by α-MSH/IBMX co-treatment. The inhibitory effects of **1a** and **1b** at 10 µM on melanin contents were similar to that of kojic acid at a concentration of 20 µM, and both compounds at 20 µM reduced melanin release to control levels.

### 3.7. Inhibitory Effect of Compounds ***1a*** and ***1b*** on Intracellular Melanin Production in B16F10 cells

We also examined the inhibitory effects of compounds **1a** and **1b** on the intracellular melanin contents of B16F10 cells (Figure 9). Intracellular melanin contents were 185% higher in α-MSH/IBMX co-treated cells than in untreated controls (100%). Treatment with **1a** or **1b** significantly and concentration-dependently reduced intracellular melanin increases induced by α-MSH/IBMX treatment, and both compounds at 10 µM more potently inhibited α-MSH/IBMX-induced increases in melanin than kojic acid at 20 µM. Surprisingly, when α-MSH/IBMX co-treated cells were treated with 20 µM of compound **1b**, intracellular melanin content was 20% lower than that of the untreated control group. These results suggest that both compounds, especially compound **1b**, more potently inhibit melanogenesis than kojic acid.

### 3.8. Inhibitory Effect of Compounds ***1a*** and ***1b*** on Cellular Tyrosinase Activities in B16F10 Cells

The intracellular inhibitory activities of compounds **1a** and **1b** against cellular tyrosinase were assessed in B16F10 melanoma cells to determine their modes of action. B16F10 cells were cultured in 6-well culture plates, exposed to four different concentrations (0, 5, 10, or 20 µM) of **1a** or **1b** for 1 h, and co-treated with IBMX and α-MSH to increase cellular tyrosinase activities. After incubation for 48 h, the optical density (OD) of each well was measured at 475 nm using a microplate reader. As the positive control, kojic acid of 20 μM was used.

Tyrosinase activity results are displayed in Figure 10. Co-treatment with IBMX and α-MSH enhanced tyrosinase activity by 91% versus untreated controls, and this increase was significantly and concentration-dependently suppressed by compounds **1a** and **1b**. At 10 µM, both compounds reduced tyrosinase activity to that achieved by 20 µM kojic acid, and at 20 µM, compound **1b** reduced tyrosinase activity to the untreated control level. Furthermore, tyrosinase activity and melanin content versus compound **1a** and **1b** concentration plots were similar, which indicates that the anti-melanogenesis effects of the two compounds were due to the inhibition of cellular tyrosinase. Although there are many examples of there being a considerable difference in the inhibitory effect of inhibitors on mushroom-derived tyrosinase and melanoma-derived tyrosinase [51,52], compounds **1a** and **1b**, which showed strong inhibition of mushroom tyrosinase, also showed potent tyrosinase inhibitory activity in B16F10 melanoma cell-derived tyrosinase.

### 3.9. Effects of Compound ***1b*** on the Expressions of Melanogenesis-Related Genes in B16F10 Cells

Since compound **1b** had by far the most potent melanogenesis inhibitory effect in B16F10 cells, we investigated whether a mechanism other than direct tyrosinase inhibition might be responsible. It has been reported that melanogenesis and cellular tyrosinase levels are positively associated, and that the immature glycosylation of tyrosinase reduces the anti-melanogenesis effect on melanoma cells [53]. Therefore, we examined the cytosolic levels of non-glycosylated tyrosinase and glycosylated tyrosinase, respectively. We observed that the ratio of glycosylated tyrosinase to total tyrosinase was significantly reduced due to **1b** in a concentration-dependent manner, which suggests that compound **1b** diminishes cellular tyrosinase activity by inhibiting tyrosinase glycosylation, and thus, melanogenesis. MITF acts as a transcription factor for *tyrosinase*, *TRP-1*, and *TRP-2* [54,55]. When we examined the nuclear MITF levels in total cell lysates by Western blotting, we found that compound **1b** significantly and dose-dependently reduced MITF protein levels at high concentrations (10–20 µM). However, the MITF protein levels increased at low concentrations (2–5 μM), indicating a conflicting result with decreased tyrosinase protein expression at the same concentrations. Due to this discrepancy, we investigated the effects of **1b** on the MITF gene targets *TRP-1*, *TRP-2*, and *tyrosinase* (Figure 11C). The mRNA levels of these genes were assessed by qRT-PCR and their mRNA expressions in compound **1b**-treated B16F10 cells and untreated controls were compared. The mRNA expression levels of the three MITF target genes were significantly and concentration-dependently suppressed by compound **1b**, implying that **1b** downregulated MITF gene expression. These results imply that the observed anti-melanogenesis effect of compound **1b** is partly due to the suppression of melanogenesis-associated genes and the inhibition of tyrosinase glycosylation.

### 3.10. DPPH Radical Scavenging Activities of Compounds ***1a*** and ***1b***

The radical scavenging effects of compounds **1a**–**1p** were investigated using a DPPH (2,2-diphenyl-1-picrylhydrazyl) assay. DPPH radical scavenging activity experiments were performed at a compound concentration of 1.0 mM, or an l-ascorbic acid (the positive control) concentration of 0.18 mM, in a methanolic solution of DPPH. Scavenging activities were assessed by measuring solution absorbances at 517 nm after a holding period of 30 min in the dark.

As shown in Figure 12, compound **1c**, which possessed a 3,4-dihydroxyphenyl moiety, exhibited the greatest DPPH radical scavenging activity (comparable to l-ascorbic acid), and six compounds (**1b**, **1d**, **1f**, **1l**, **1m**, and **1n**) with at least one hydroxyl substituent on the phenyl ring exhibited moderate DPPH radical scavenging efficacy. The other nine compounds, which included compounds bearing no hydroxyl substituent on the phenyl ring, showed no or weak radical scavenging activities. These results demonstrate that there is a relationship between DPPH radical scavenging activity and the number of hydroxyl substituents on the phenyl ring. In addition, the position of the hydroxyl on the phenyl ring influenced DPPH radical scavenging activity. For example, compound **1b**, which most potently inhibited tyrosinase, had two hydroxyl substituents on the phenyl ring, equivalent to compound **1c**, however, unlike compound **1c**, it had only a weak DPPH radical scavenging effect. 

### 3.11. The ABTS Radical Scavenging Effects of Compounds ***1a***–***1p***

ABTS radical scavenging effects were investigated during antioxidant evaluations. Radical scavenging activities were assessed by measuring absorbances of the ABTS radical cation at 734 nm after exposure to compounds **1a**–**1p** (100 µM) in the dark for 2 min. Trolox was used as the positive reference control. ABTS free radical scavenging results are shown in Figure 13.

Compounds **1b**, **1c**, and **1n** potently scavenged the ABTS radical, and **1c**, bearing a catechol moiety, was the most potent, which concurs with our DPPH free radical scavenging results. Compound **1c** and trolox scavenged 92.4% and 98.9%, respectively, of ABTS free radical activity. Notably, compound **1b**, which had the most potent anti-melanogenesis effect, also potently scavenged ABTS radical activity (75% inhibition), but only weakly scavenged the DPPH radical. Compounds **1d**, **1e**, **1g**, **1l**, and **1m** exhibited moderate ABTS radical scavenging activities, ranging from 30.9 to 50.0%. As was observed for DPPH radical scavenging activities, compounds bearing no hydroxyl substituent on the phenyl ring showed no or weak ABTS radical scavenging activities. Compound ABTS and DPPH radical scavenging activity patterns were similar, except for compound **1b**.

### 3.12. The Intracellular ROS Scavenging Effects of the 16 2-Arylbenzothiazole Derivatives

A report that ROS might regulate melanogenesis in melanoma cells [56] led us to investigate the scavenging effects of compounds **1a**–**1p** on intracellular ROS. A DCFH-DA (2′,7′-dichlorodihydrofluorescein diacetate) assay was used to measure the abilities of compounds **1a**–**1p** to scavenge intracellular ROS. DCFH-DA spreads across cell membranes and is hydrolyzed to DCFH (2′,7′-dichlorodihydrofluorescein), and DCFH reacts with ROS to form DCF (2′,7′-dichlorofluorescein) [57].

The intracellular ROS scavenging activities of compounds **1a**–**1p** were evaluated in vitro using DCFH-DA, and 3-morpholinosydnonimine (SIN-1) was used as a spontaneous RNS/ROS generator [58,59]. Treatment of B16F10 cells with 10 µM SIN-1 greatly increased intracellular ROS generation (Figure 14). With the exception of compound **1j**, all compounds at 20 µM significantly scavenged intracellular ROS, and six compounds, viz. **1a**–**1c**, **1g**, **1n**, and **1p**, scavenged intracellular ROS to a similar extent to trolox (20 µM; the positive control). In particular, compound **1c**, which possessed a catechol moiety and most potently scavenged DPPH and ABTS radicals, scavenged intracellular ROS far more potently than trolox. This observation that compounds possessing the catechol moiety exhibit strong intracellular ROS scavenging activities concurs with previous reports [60,61].

## 4. Conclusions

During our ongoing search for potent tyrosinase inhibitors, 16 2-arylbenzothiazole compounds were synthesized using one-step reactions. Compound **1b** had an IC_50_ value of 0.2 ± 0.01 μM against mushroom tyrosinase, and Lineweaver–Burk plots indicate that **1b** competitively inhibited tyrosinase in the presence of l-tyrosine or l-DOPA. Docking simulation results indicate that the presence of hydroxyl at the 2 and 4 positions of the phenyl ring of **1b** markedly enhanced tyrosinase inhibitory activity by coordinating and forming a salt bridge with the zinc ions of tyrosinase and one hydrogen bond. Results from an in vitro assay using B16F10 cells suggest that **1a** and **1b** exert strong anti-melanogenic effects by inhibiting intracellular tyrosinase inhibitory activity without perceptible cytotoxicity. In addition, the potent anti-melanogenic effect of compound **1b** was partially attributable to the inhibition of tyrosinase glycosylation and the expression of melanogenesis-related genes (*tyrosinase*, *TRP-1*, and *TRP-2*). Furthermore, compound **1b** strongly scavenged ROS and ABTS radicals. These results indicate that compound **1b** may be considered a developmental lead compound with potent anti-tyrosinase and antioxidant activities.

## Data Availability

Data are contained within the article and Appendix A.

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
