# Peer review of "A Novel Class of Potent Anti-Tyrosinase Compounds with Antioxidant Activity, 2-(Substituted phenyl)-5-(trifluoromethyl)benzo[*d*]thiazoles: In Vitro and In Silico Insights"

_antioxidants, 2022, doi:10.3390/antiox11071375_

Round 1

Reviewer 1 Report

The work is very intense since organic syntheses are complex. On the other hand, the study of the synthesized compounds, as inhibitors of the enzyme, is not entirely clear, for example: in table 3, IC50 values ​​are shown for compound 1b in the monophenolase and diphenolase activities. These values ​​are different for the two activities, when the inhibitor is said by the authors to be competitive. This would require that the KI values ​​of the two activities be the same and thus the IC50's would also be the same. In the same table, kojic acid, which is a classic tyrosinase inhibitor, shows the same IC50 values, could this discrepancy between KI and IC50 values ​​be due to the fact that this compound is an alternative tyrosinase substrate?

Author Response

The work is very intense since organic syntheses are complex. On the other hand, the study of the synthesized compounds, as inhibitors of the enzyme, is not entirely clear, for example: in table 3, IC50 values ​​are shown for compound 1b in the monophenolase and diphenolase activities. These values ​​are different for the two activities, when the inhibitor is said by the authors to be competitive. This would require that the KI values ​​of the two activities be the same and thus the IC50's would also be the same. In the same table, kojic acid, which is a classic tyrosinase inhibitor, shows the same IC50 values, could this discrepancy between KI and IC50 values ​​be due to the fact that this compound is an alternative tyrosinase substrate?

Thank you for your valuable comments. First, we have checked previously reported articles. Reference 1 below shows that reversible competitive inhibitors can have different IC50 values in monophenolase IC50 values and in diphenolase IC50 values. In addition, references 2 and 3 show that compounds acting as competitive inhibitors in both mono and diphenolase activity can have different IC50 values and different Ki values in the presence of L-tyrosine and L-Dopa. These compounds, like compound 1b in our manuscript, commonly have a 4-hydroxyphenyl group in their structure. Therefore, we think that the possibility of alternative tyrosinase substrates in 1b will be related to different Ki values. We plan to study experiments to prove whether compound 1b acts as an alternative tyrosinase substrate. It is thought that a long study time will be required, and the results will be published in the future. In addition, we also think that the lag time (ref. 4) arising from monophenolase activity may be related to different Ki values. Thanks to the reviewer's comments, I learned more about tyrosinase. Thank you again for the critical comment.

# Reference 1: Structure–activity relationships of anti-tyrosinase and antioxidant activities of cinnamic acid and its derivatives. Bioscience, Biotechnology, and Biochemistry, 2021, Vol. 85, No. 7, 1697-1705.

# Reference 2: Discovery of a Highly Potent Tyrosinase Inhibitor, Luteolin 5‑O‑β‑D‑Glucopyranoside, Isolated from Cirsium japonicum var. maackii (Maxim.) Matsum., Korean Thistle: Kinetics and Computational Molecular Docking Simulation. ACS Omega 2018, 3, 17236−17245

# Reference 3: Tyrosinase Inhibition and Kinetic Details of Puerol A Having But-2-Enolide Structure from Amorpha fruticose. Molecules 2020, 25, 2344.

# Reference 4: Inhibitory Effects of 4-Halobenzoic Acids on the Diphenolase and Monophenolase Activity of Mushroom Tyrosinase. The Protein Journal, Vol. 23, No. 5, July 2004

Reviewer 2 Report

The work from the point of view of organic synthesis, in my opinion I think it is correct. However, in the kinetic study of tyrosinase inhibition, it must be taken into account that the compounds that act as potent inhibitors carry a free OH in the R3 position (Table 2). This structure would allow tyrosinase to be considered as an alternative substrate to L- tyrosine and L-dopa (Compounds 1a and 1b), compound 1c has an o-diphenolic structure and, therefore, can be a substrate for the enzyme.

The authors should show that compounds 1a, 1b and 1c are not substrates of tyrosinase, since if they behave as substrates, the kinetic analysis is invalid.

Author Response

The work from the point of view of organic synthesis, in my opinion I think it is correct. However, in the kinetic study of tyrosinase inhibition, it must be taken into account that the compounds that act as potent inhibitors carry a free OH in the R3 position (Table 2). This structure would allow tyrosinase to be considered as an alternative substrate to L- tyrosine and L-dopa (Compounds 1a and 1b), compound 1c has an o-diphenolic structure and, therefore, can be a substrate for the enzyme.

The authors should show that compounds 1a, 1b and 1c are not substrates of tyrosinase, since if they behave as substrates, the kinetic analysis is invalid.

Thank you for your valuable comments.

We conducted the kinetic study in the presence of L-tyrosine or L-DOPA (1, 2, 4, 8, and 16 mM), using compound 1b (0, 0.1 0.2, and 0.4 µM in L-tyrosine, and 0, 1, 2, and 4 µM in L-DOPA). Experiments were performed at substrate concentrations that were at least 2,500 – 40,000-fold (in L-tyrosine) or 250 – 4,000-fold higher (in L-DOPA) compared to the inhibitor (1b). Therefore, it does not make sense that 1b is an alternative substrate that are thousands to tens of thousands of times better than native substrates (L-tyrosine and L-DOPA). Although unlikely, it is possible that 1b is an alternative substrate, as the reviewer noted. Therefore, we plan to study experiments to prove this. It is thought that a long study time will be required, and the results will be published in the future. Thank you again for the critical comment.

Reviewer 3 Report

In this manuscript, authors describe synthesis of benzothiazole derivatives, and their inhibitory effects on mushroom tyrosinase, cell-based assay, etc.

Many bioassays were performed, and the results are partly interesting, but the following points should be revised.

Major:

It has been reported that there is a considerable difference in the inhibitory effects of inhibitors on mushroom-derived tyrosinase and melanoma-derived tyrosinase.

Authors should cite the related references, and the differences should be described in the introduction and discussion.

Radial scavenging activity of test compounds may contribute to the inhibition of melanogenesis. Their contribution ratio? Does radical scavengers inhibit the melanogenesis?

Minor:

2.2.7. Characterization of compounds 1c-lp

‘Solid’ is not suitable. Crystal?

J values should be same to that of corresponding protons.

It is better that effective figure numbers of inhibition (%) and IC50 should be rounded.

For example: 30.37±0.31 --> 30.4±0.31     54.24 µM --> 54.2 µM

Compound number should be bold face.

Author Response

In this manuscript, authors describe synthesis of benzothiazole derivatives, and their inhibitory effects on mushroom tyrosinase, cell-based assay, etc.

Many bioassays were performed, and the results are partly interesting, but the following points should be revised.

Major:

It has been reported that there is a considerable difference in the inhibitory effects of inhibitors on mushroom-derived tyrosinase and melanoma-derived tyrosinase.

Authors should cite the related references, and the differences should be described in the introduction and discussion.

Thank you for your valuable comments. We have briefly described differences between mammalian tyrosinases and mushroom tyrosinase in the ‘Introduction’, along with a related reference. In addition, we have briefly described that there is a considerable difference in the inhibitory effects of inhibitors on mushroom-derived tyrosinase and melanoma-derived tyrosinase in the ‘Section 3.8.’ and have cited the related references.

Radial scavenging activity of test compounds may contribute to the inhibition of melanogenesis. Their contribution ratio? Does radical scavengers inhibit the melanogenesis?

Thank you for your valuable comments. High ROS production and high oxidative stress levels, including radical species, can increase melanogenesis. Therefore, radical scavengers have the potential to inhibit melanogenesis (refs 1 – 3). It is impossible for us to say how much the radical scavenging activity of the test compound contributes to the inhibition of melanogenesis, but the contribution of the radical scavenging activity to the inhibition of melanogenesis seems to be weaker than the contribution of direct tyrosinase inhibition.

# Reference 1: Novel Chemically Modified Curcumin (CMC) Derivatives Inhibit Tyrosinase Activity and Melanin Synthesis in B16F10 Mouse Melanoma Cells. Biomolecules 2021, 11, 674.

# Reference 2: ROS as Regulators of Cellular Processes in Melanoma. Oxidative Medicine and Cellular Longevity Volume 2021, Article ID 1208690.

# Reference 3: Melanocytes as Instigators and Victims of Oxidative Stress. J. Investig. Dermatol. 2014, 134, 1512–1518.

Minor:

2.2.7. Characterization of compounds 1c-lp

‘Solid’ is not suitable. Crystal?

→ Thank you for your comment. We have deleted the word ‘Solid’ because the properties of compounds synthesized were already described in General synthetic procedure and in the respective synthetic procedures for the other compounds.

J values should be same to that of corresponding protons.

→ Each J value was made to match the J value of the corresponding proton, as suggested by reviewers.

It is better that effective figure numbers of inhibition (%) and IC50 should be rounded.

For example: 30.37±0.31 --> 30.4±0.31     54.24 µM --> 54.2 µM

→ Valid figures for % and IC50 values have been rounded off as suggested by reviewers.

Compound number should be bold face.

→ All compound numbers have been corrected in bold.

Thank you again for valuable comments.

Round 2

Reviewer 1 Report

The explanations given by the authors seem sufficient to me

Reviewer 2 Report

I agree with what the authors have answered.

Reviewer 3 Report

This manuscript is well improved according to the suggestions by reviewers.